# MSSI-Based Dispersion-Managed Link Configured by Randomly-Distributed RDPS Only in Former Half Section

Jae-Pil Chung [1] and Seong-Real Lee [2,*]

1   Department of Electronic Engineering, Gachon University, Seongnam 13120, Korea
2   Division of Navigational Information System, Mokpo National Maritime University, Mokpo 58628, Korea
*   Correspondence: reallee@mmu.ac.kr

**Abstract:** The weakness of the dispersion-managed link, which is combined with optical phase conjugation to compensate for optical signal distortion caused by chromatic dispersion and the nonlinear Kerr effect of the standard single mode fiber is, its limited structural flexibility. We propose a dispersion map that can simultaneously compensate for the distorted wavelength division multiplexed signal while increasing the configurational flexibility. Each residual dispersion per span (RDPS) in the former half of the proposed link is randomly determined, and in the latter half, the arrangement order of RDPS is the same as or inverted in the former half. We confirm that the dispersion maps in which the RDPS distribution pattern in the latter half is opposite to the arrangement order in the former half are more effective in compensation, and the compensation effect is better than in the dispersion map of the conventional scheme. The notable result of this study is that the flexibility can be increased by randomly arranging RDPS in the former half, and compensation improvement can be achieved by inversing the order of RDPS arrangement of the former half in the latter half, which makes the dispersion profile of each half link roughly symmetric with respect to the midway optical phase conjugator.

**Keywords:** dispersion management; mid-span spectral inversion; dispersion map; optical phase conjugator; residual dispersion per span; random distribution; chromatic dispersion; nonlinear Kerr effect; wavelength division multiplexed

## 1. Introduction

In optical links consisting of a standard single mode fiber (SSMF), chromatic dispersion and the nonlinear Kerr effect are the significant intrinsic barriers to increasing transmission distance and bit rate. Optical phase conjugation has proved to be an efficient technique to compensate for the signal impairment due to both phenomena. Principally, the optical phase conjugator (OPC) is placed in the middle of the total transmission length, and this scheme is referred to as a mid-span spectral inversion (MSSI). In this system, the entire signal is distorted through the former half of the total link preceding the midway OPC being phase conjugated. The dispersive and nonlinear Kerr effect of the phase-conjugated signal are reversed through the latter half of the total link, and finally the deteriorated signal can be recovered, if the dispersion-power profile is symmetric with respect to the OPC position [1,2]. The symmetry of dispersion-power profile is difficult to establish in a real link because of loss of fibers and amplification of signal power by erbium-doped fiber amplifiers (EDFAs) in a lumped amplification system [3].

Several approaches to maximize the symmetry in MSSI have been proposed. Among of them, applying short span length to the transmission link contributes to increasing the symmetry [2]; however, this increases the required number of EDFAs, and consequently the nonlinear Kerr effect is further increased as optical signal power increases. Other approaches include the use of distributed Raman amplification [4,5] and applying MSSI to the dispersion-managed link [6,7]. Another limitation of optical phase conjugation in a real

link is the difficulty of implementing a flexible link configuration. That is, the transmission link parameters, such as the length of the SSMF, dispersion accumulated in each span, and more importantly the OPC position have to be determined to symmetrize the local dispersion distribution and power profile with respect to the OPC. For this reason, the OPC position should be fixed around midway of the total link, the length of each SSMF should be generally constant, and maximum launch power of signal is restricted. Thus, the variety of the link configuration is limited in the optical phase conjugation system for compensation of chromatic dispersion and the nonlinear Kerr effect.

Dispersion management (DM) is a well-known approach that can mitigate fiber chromatic dispersion [8–11]. The dispersion coefficient of the dispersion-compensating fiber (DCF) is opposite to the SSMF; thus, the dispersion accumulated in the SSMF can be eliminated or reduced by controlling the length and coefficient of the DCF. In the DM link, if Kerr nonlinearity is not present and input power is sufficient to overcome amplified spontaneous emission (ASE) [12] noise impairment, any bit rate can be transmitted for any distance.

In the DM link configuration, the significant link parameter capable of impacting the dispersion compensation is residual dispersion per span (RDPS), which is defined as dispersion accumulated in each fiber span. The RDPS is determined by the lengths and dispersion coefficients of the SSMF and DCF. The simplest configuration in the DM link is where the same RDPS is applied to each fiber span of whole link. This configuration means the lengths and coefficients of fibers are fixed to uniform values in every span. However, this uniform distribution of RDPSs is also restricted to adaptive implementation of the DM link required by the demand of network design plan.

One approach to supplement the incomplete compensation due to the difficulty of symmetry in MSSI is in combination of DM link configurations [6,7]. Authors have also confirmed a compensation improvement of 960 Gb/s wavelength division multiplexed (WDM) transmission in an MSSI-based DM link [13–15]. In our studies related to applying MSSI to the DM link, various DM configurations have been proposed by using non-uniform RDPSs, such as artificial distribution of arbitrary RDPSs, random distribution of various RDPSs, and an asymmetric dispersion map with respect to the midway OPC, to expand flexibility of the link configuration. On the flexibility front, the best scheme is random distribution, i.e., RDPSs with different magnitude are freely allocated for each fiber span, with no intention of deployment. However, the random distribution of RDPSs results in unsatisfactory compensation, since it is difficult to symmetrize the dispersion-power profile with respect to the midway OPC. From the viewpoint of performance, on the other hand, the best compensation is achieved by artificial distribution and predetermined RDPSs, especially the gradually ascending or descending distribution of these RDPSs in each half link. The symmetry of the dispersion profile of each half is ultimately maintained with respect to the midway OPC.

From analysis of our previous studies, it was confirmed that there is a conflict of interest between the conditions of link parameters for flexibility and for the compensation effect. That is, it is difficult to simultaneously achieve the two aims of flexibility and the best compensation through the established attempts using RDPS distributions. Therefore, another method is required to increase the compensation performance and to extend flexibility. A possible and simple approach is the compromise method, i.e., the selective mixing of various distributions of RDPSs for each half link.

In this study, we numerically demonstrated the compensation of the 960 Gb/s WDM signal in the MSSI-based DM link, in which RDPSs of only the former half link were randomly distributed, and RDPSs of the remaining half link were artificially distributed. The two methods of deployment of RDPSs in the latter half link were as follows: reversing or following those in the former half link. We also assessed the compensation performance in the DM link configured by all-randomly distributed RDPSs in both halves for comparison with the proposed schemes.

## 2. Dispersion-Managed Link and WDM System Modeling

### 2.1. Dispersion-Managed Link

The dispersion-managed link with the embedded midway OPC for transmitting WDM 24-channels of 40 Gb/s is illustrated in Figure 1. Each half link comprises 27 fiber spans; thus, total link consists of 54 fiber spans. All fiber spans include the SSMF and DCF, and the length of the SSMF is fixed to 80 km. The previous studies have shown that the deployment of fibers in the fiber span affects the compensation of each distorted channel [13–15]. The compensation is further increased when the arrangements of the SSMF and DCF are opposite with respect to the OPC. In this study, the DCF preceded the SSMF in every fiber span of the former half link; the DCF, on the other hand, was placed after the SSMF in the latter half link.

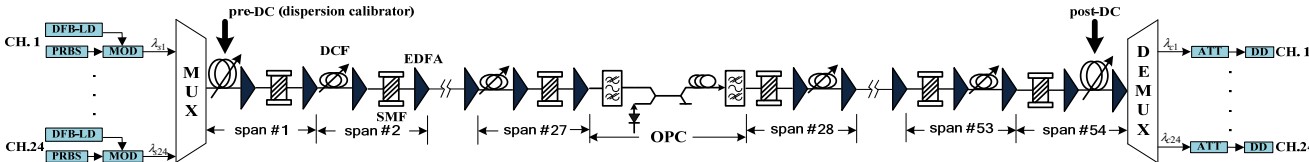

**Figure 1.** The 960 Gb/s WDM transmission system through the dispersion-managed link and the midway OPC.

In this study, the RDPS of fiber span was randomly selected as 1 of 27 values from $-1300$ to 1300 ps/nm, which were distinguished with intervals of 100 ps/nm. Each RDPS was determined by the length of the DCF because the length of the SSMF and the dispersion coefficients of the SSMF and DCF are fixed. We assumed the SSMF parameters as follows: the dispersion coefficient was 17 ps/nm/km, the attenuation coefficient was 0.2 dB/km, and the nonlinear coefficient was 1.41 $W^{-1}$ $km^{-1}$. The DCF was characterized as follows: the dispersion coefficient was $-100$ ps/nm/km, the attenuation coefficient was 0.6 dB/km, and the nonlinear coefficient was 5.06 $W^{-1}$ $km^{-1}$. These parameter values were assumed for wavelength of 1550 nm.

The completed random distribution of RDPSs means the arrangement order of random-value RDPS is not fixed in both halves. However, as mentioned earlier, the MSSI-based DM link configured with completed random distribution does not sufficiently compensate for the distorted WDM channels because it is difficult to establish the symmetry of local dispersion with respect to the OPC. Therefore, in this work, the RDPS distribution in the former half link is only essentially random, while the distribution in the latter half link is two cases with artificially arranged RDPSs as follows. The first case is expressed as "random-inverse", in which the RDPS of each fiber span in the former half link is randomly and independently determined as 1 of 27 values, while the arrangement of RDPSs for each fiber span in latter half is opposite to that of the former half. For the second case, in the latter half, the arrangement of the RDPSs of the fiber span is the same as the randomly aligned RDPSs in the former half; this case is called "random-follow". We also considered "all-random" distribution, in which the RDPS of each fiber span is simultaneously randomly arranged in both halves to compare the compensation performance.

In considering three cases, we investigated 50 different distribution patterns. However, for a reasonable analysis of the compensation performance, the RDPS arrangement of the former half link was kept the same in all three cases. In summary, after randomly arranging 27 RDPSs for each fiber span of the former half, the RDPS for each fiber span in latter half was allocated in the same order as the arrangement in the former half (random-follow distribution), or the arrangement order of RDPS in the latter half was reversed with the former half (random-inverse distribution). Dispersion-managed links with completely different random arrangement of RDPS in the former half and latter half were also considered. These processes were independently repeated 50 times.

It is well known that the best compensation occurs when the net residual dispersion (NRD) is sustained near but not 0 ps/nm in "pseudolinear" systems [16]. Our investigated

system is also included the pseudolinear system. Thus, the arbitrary fiber span should control the NRD to optimal value. The first span and the last span play this role, and they are called the pre-dispersion calibrator (DC) and post-DC, respectively. However, in this research, only the pre-DC play the role of NRD controller of the total link by varying its own length; that is, the DCF length in the first span is varied to decide the NRD of the former half link, and simultaneously, the DCF length of the last span is fixed to make the NRD of the latter half link 0 ps/nm.

### 2.2. WDM Transmitters, Receivers, and Midway OPC

The transmission part of each WDM channel consists of a pseudorandom bit sequence (PRBS) for data generation, a distributed feedback laser diode (DFB-LD) for light source, and an external modulator. First, an independent 40 Gb/s 127 (=$2^7 - 1$) PRBS is generated and then modulates the intensity of the light emitted by DFB-LD in the external modulator. Following ITU-T recommendation G.694.1, the center wavelength of each DFB-LD is assumed to be from 1550 to 1568.4 nm with 100 GHz (0.8 nm) intervals. The output of the modulator is assumed to be return-to-zero (RZ) pulses. The output electric field of the RZ format is assumed to be chirp-free with a second-order super-Gaussian pulse with a 10 dB extinction ratio (ER) and a duty cycle of 0.5.

The RZ pulses of 24 channels are multiplexed in the multiplexer (MUX) and propagated through the former half of the dispersion-managed link. However, the propagated signals are deteriorated by the chromatic dispersion and Kerr nonlinearities. The distorted optical signals are phase-conjugated in the midway OPC through the four-wave mixing (FWM) processes of the input waves and the pump light. Highly nonlinear dispersion-shifted fiber (HNL-DSF) was selected as a nonlinear medium for phase conjugation. The power and wavelength of the pump light for generating the phase-conjugated waves were assumed to be 18.5 dBm and 1549.75 nm, respectively. Consequently, each wavelength of the conjugated 24 channels in the midway OPC was then arranged from 1549.5 to 1528.5 nm (−0.8 nm intervals).

The conjugated WDM signals are propagated through the remaining half of the total link and then demultiplexed. We assumed that each receiver follows the direct detection method. The receiver bandwidth was assumed to be 0.65 times 40 Gb/s. The WDM receiver comprises an EDFA with a noise figure of 5 dB, an optical filter, a PIN diode as the photodetector, a pulse-shaping Butterworth filter, and a decision circuit.

### 3. Simulation Method and Numerical Assessment

The nonlinear Schrödinger equation (NLSE) of Equation (1) represents the propagation of the optical signal in the silica fibers, such as the SSMF and DCF, where chromatic dispersion and nonlinear Kerr effect dominate [17].

$$\frac{\partial A_j}{\partial z} = -\frac{\alpha}{2}A_j - \frac{i}{2}\beta_{2j}\frac{\partial^2 A_j}{\partial T^2} + \frac{1}{6}\beta_{3j}\frac{\partial^3 A_j}{\partial T^3} + i\gamma_j|A_j|^2 A_j + 2i\gamma_j|A_k|^2 A_j, \tag{1}$$

In Equation (1), $j$, $k$ = 1, 2, . . . , 24 ($j \neq k$), $A_j$ represents the complex amplitude of the optical signal of the $j$-th channel, $z$ is the propagation distance, $\beta_{2j}$ is the group velocity dispersion (GVD), $\beta_{3j}$ is the dispersion slope, $\gamma_j$ is the nonlinear coefficient, and $T = t - z/v_j$ is the retarded frame. The last two terms of (1) express the effect of nonlinearity including self-phase modulation (SPM) and cross-phase modulation (XPM). The XPM has little impact on the system performance because of the high dispersion coefficient of the SSMF and, hence, its high walk-off [18]. Thus, the XPM effect on dense WDM signals was omitted in our simulation. The most common approach for solving (1) is the split-step Fourier (SSF) method [17].

The SSF method divides the transmission medium dominated by NLSE into small steps and processes linear calculations in the frequency domain and nonlinear calculations in the time domain separately and continuously through Fourier transform and inverse Fourier transform for each step [17]. In this study, the DM link was modeled by applying

the SSF method at intervals of 0.1 km. The multiplexed signal with RZ pulses having different wavelengths as described above is propagated through the DM link modeled in this way. The demultiplexed signal was recovered using the direct detection method for each channel. All coding was carried out using MATLAB.

$$EOP\ [dB] = 10\ log_{10} \frac{EO_{rec}}{EO_{btb}}, \tag{2}$$

We used the eye opening penalty (EOP) and timing jitter of the received optical signal as the assessment factor. The EOP is expressed by Equation (2), where $EO_{rec}$ and $EO_{btb}$ are the eye opening of receiving optical signals and the back-to-back optical signals (i.e., the reference signals), respectively. The eye opening is defined as twice the averaged power of all of the received signals divided by the smallest power level of the "one" optical signal minus the largest power level of the "zero" optical signal.

The compensation performance criterion is a 1-dB EOP, which is equivalent to the pulse broadening (the ratio of root mean square (RMS) width of the received pulse to RMS width of the initial pulse) of 1.25 and corresponds to a bit error rate (BER) of $10^{-12}$ [19]. We also used 2.5 ps, which is one-tenth of the duration of 40 Gb/s, as the performance criterion for timing jitter.

## 4. Simulation Results and Discussion

Figure 2 shows the eye diagrams of the worst channel launched with 5 dBm power into the dispersion-managed link configured by the three random cases and the conventional scheme. The conventional scheme of Figure 2a corresponds to the case where the RDPS of all fiber spans is uniformly distributed by 0 ps/nm. For comparison, the results of Figure 2b–d were obtained by making the RDPS arrangement order of each fiber span in the former half the same as the following: 1100, 900, −600, 800, 1200, −1000, −400, 300, 100, −900, 0, −100, −500, −1200, −1300, −1100, −700, 400, 700, 1300, −300, 600, 1000, −200, 200, 500, and −800 ps/nm.

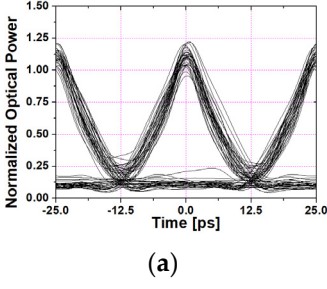 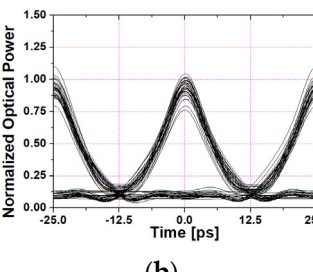 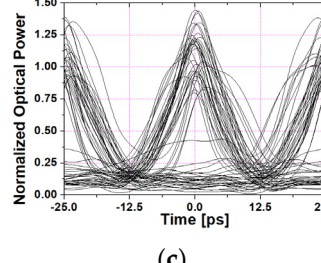 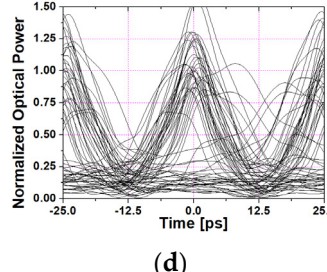

(**a**)  (**b**)  (**c**)  (**d**)

**Figure 2.** The eye diagrams. (**a**) conventional scheme, (**b**) random-inverse distribution, (**c**) random-follow distribution, and (**d**) all-random distribution.

By comparing Figure 2b–d, it can be seen that the reception quality is excellent in the dispersion-managed link with the random-inverse distribution shown in Figure 2b. The compensation result in this case is an improvement on the conventional scheme of Figure 2a. These results are considered to be related to the basic condition for compensation through the midway OPC. In other words, it can be expected that the compensation of distorted channels through MSSI increases as the dispersion distribution becomes more symmetric around the midway OPC, and random-inverse corresponds to the distribution close to the symmetry.

Unlike Figure 2a,b, the received pulse sequence of Figure 2c,d shows very severe amplitude fluctuations and phase fluctuations (i.e., timing jitter). In a pseudolinear system where chromatic dispersion and SPM of the nonlinear Kerr effect exist, optical pulse distortion occurs through the interaction of the two phenomena. That is, when the amplitude (intensity) of the optical pulse is changed due to chromatic dispersion, the nonlinear phase

shift is generated according to the changed intensity, resulting in distortion of the optical pulse. Applying this mechanism to the eye diagram analysis shown in Figure 2, the eye diagrams in Figure 2c,d show that the specific random-follow and all-random distributions do not sufficiently compensate for the amplitude and phase distortion caused by the interaction between the chromatic dispersion and SPM experienced by the optical pulse. On the other hand, when the random-inverse distribution is applied, it can be confirmed that the mitigation of the amplitude and phase fluctuations of the optical signal due to the interaction of chromatic dispersion and SPM is effective.

Since the EOP in the optical domain is the result of measuring the comprehensive eye closure of the optical pulse due to the amplitude change from the various causes, it is possible to analyze the compensation of the signal distortion due to SPM and chromatic dispersion by using the EOP. Thus, the compensation performance was analyzed through the EOP and, if necessary, timing jitter was also used.

In the conventional scheme, the maximum launch power values that produce a 1-dB EOP and 2.5 ps of timing jitter were 4.13 and −0.13 dBm, respectively. Figure 3a,b show the maximum launch power values corresponding to the 1-dB EOP and 2.5 ps timing jitter, respectively. In Figure 3a,b, "descending order number" means that the resulting maximum launch power is sorted from the largest value to the smallest value, in order to facilitate comparative analysis.

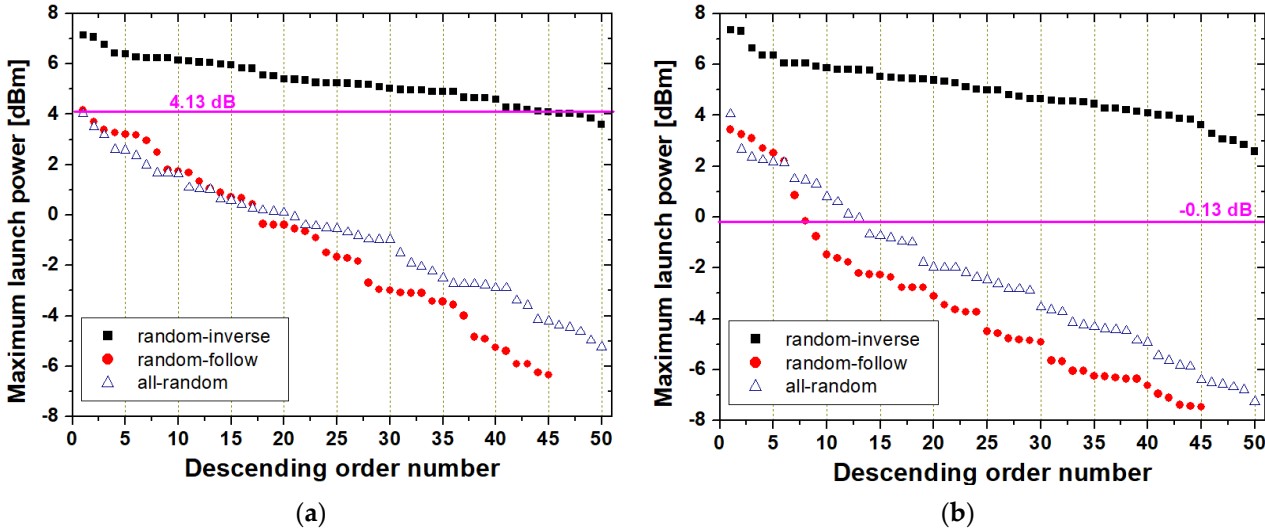

(a)

(b)

**Figure 3.** The maximum launch power of the worst channel. (**a**) Resulting 1-dB EOP, (**b**) resulting 2.5 ps timing jitter.

First, it can be seen from the result of Figure 3a that the EOP characteristics of the random-inverse distribution are overall better than the conventional scheme except for three cases. In particular, in timing jitter analysis, it can be confirmed from Figure 3b that the compensation characteristics in all 50 random-inverse distributions are superior to those of the conventional scheme. Notably, in Figure 3a,b, the compensation characteristics in the all-random distribution are generally better than those in the random-follow. This result suggests that the case number of making the dispersion distribution close to symmetric with respect to the midway OPC is greater in the all-random distribution than in the random-follow distribution. In contrast, the random-follow distribution deviates significantly from symmetry because the dispersion map of the latter half is the same as that of the former half.

In Figure 3, "descending order number" is used for convenience of comparison, but all of three random distribution cases were performed 50 times with different random arrays. The pattern numbers were assigned in the order in which simulations were performed. Figure 4 compares pattern number 48 with the largest difference in performance between the EOP and timing jitter and pattern number 49 with the smallest difference in random-inverse

distribution. It can be seen from Figure 4 that, depending on the random distribution pattern of RDPS, the variation of timing jitter is larger than that of the EOP. This can be confirmed in Figure 4, where the difference in the launch power, which produces 2.5 ps timing jitter, is larger than the difference in launch power that produces a 1-dB EOP.

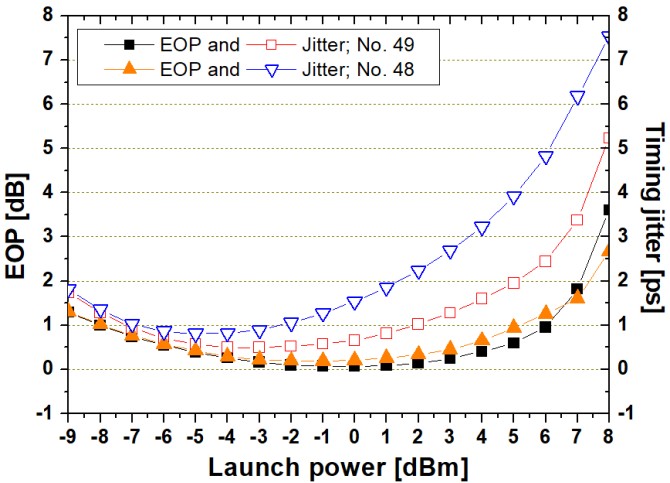

**Figure 4.** The EOP and timing jitter of pattern number 49 and 48 as a function of the launch power in case of random-inverse distribution.

The flexibility is included in the quality evaluation factor for the design of optical transmission links. The effective NRD range can be used as a measure to evaluate the flexibility of the dispersion-managed link. In the analysis, the NRD was fixed at 10 ps/nm because this value can induce the best compensation. However, it was confirmed through the previous studies that NRD can obtain the valid compensation even if it has value larger or smaller than 10 ps/nm. Thus, the effective NRD range can be defined as the range from the minimum NRD to the maximum NRD at which the 1-dB EOP can be achieved.

Figure 5 shows the effective NRD range in each random distribution for which the best compensation is obtained. To be more specific, the results obtained for pattern number 28 in random-inverse distribution and pattern number 48 in random-follow and all-random distributions are represented in Figure 5. As can be seen intuitively, the effective NRD range shown in Figure 5 has a closed curve shape. It can be seen that the larger the area of the closed curve, the greater the flexibility of the link design. Consistent with the results obtained above, it can be confirmed that the flexibility of the dispersion-managed link design is excellent and is in the order of random-inverse, all-random, and random-follow.

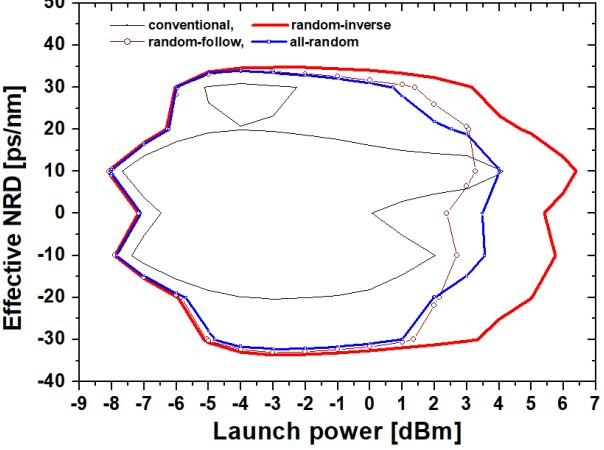

**Figure 5.** The effective NRD ranges versus the launch power.

In terms of performance, the area of the closed curve, i.e., the area of the effective NRD range, is equivalent to the product of NRD and launch power. We evaluated the effective NRD range for a total of 150 RDPS distribution patterns. However, it is difficult to express and analyze the effective NRD range with one graph, since the number of data is too large. An easy and convenient way to compare flexibility is to use the product of NRD and launch power for each random pattern.

Figure 6 shows the product of NRD and launch power for all distributions investigated in this study. It can be confirmed that random-inverse distribution is the best in terms of the flexibility of the dispersion-managed link design. Moreover, it can be seen that the product of NRD and launch power for all random-inverse distribution patterns is larger than the conventional scheme (333.6 (ps/nm)·dBm).

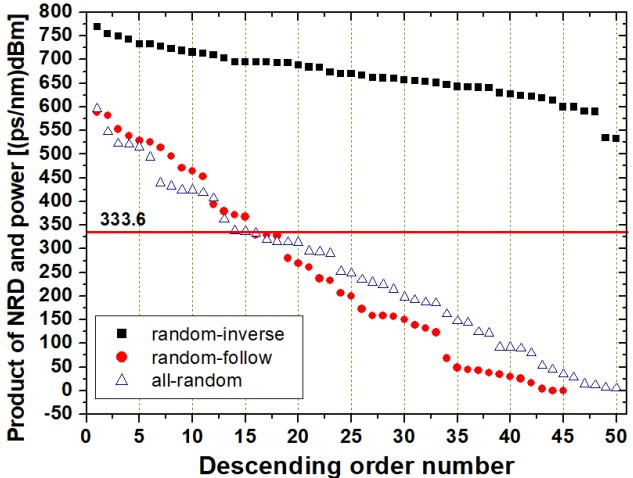

**Figure 6.** The product of NRD and launch power.

As a result of the analysis of the product of NRD and launch power, it can be confirmed that the effective NRD range and the maximum launch power resulting in a 1-dB EOP may not be simultaneously large even if the product is larger. That is, the design margin of the NRD and the launch power varies depending on the specific random pattern of the RDPS. Figure 7 shows the product and the maximum launch power that produces a 1-dB EOP for each pattern.

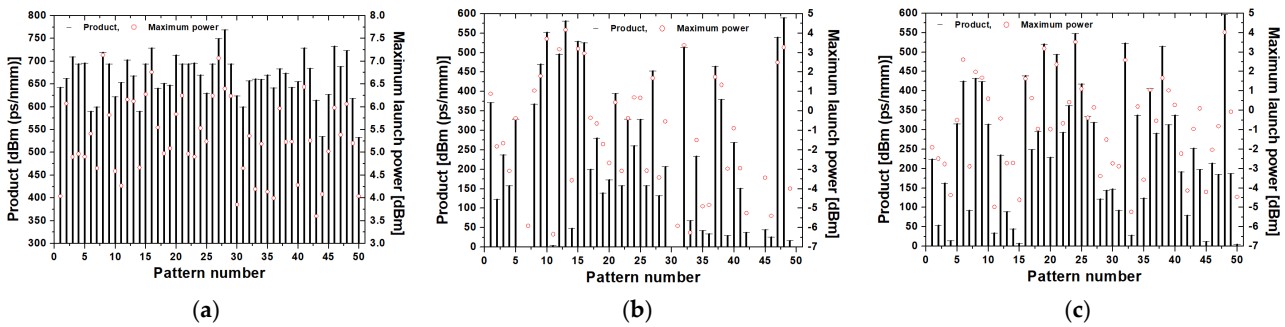

**Figure 7.** The product of NRD and launch power and the maximum launch power, which result in 1-dB EOP. (**a**) Random-inverse case, (**b**) random-follow case, and (**c**) all-random case.

In the result of Figure 7a, the maximum launch power for pattern numbers 33 and 34 with similar product magnitudes are 4.2 and 5.2 dBm, respectively, with a difference of about 1-dB. This result means that in the dispersion-managed link designed with pattern number 33, the NRD can be generously selected as the maximum launch power and is lower compared to pattern number 34. Among the random-inverse distributions, the best product characteristic is obtained by applying pattern number 28, and since the

maximum launch power is not too large, NRD can be applied with sufficient margin. On the other hand, in the dispersion-managed link designed by pattern number 8, although the product characteristic is relatively good, it is difficult to apply NRD with the margin due to the relatively large maximum launch power. In conclusion, the RDPS pattern in the random-inverse distribution should be determined and applied according to the specific requirements such as NRD margin and power margin.

Figure 8a shows the dispersion maps made with the RDPS distribution pattern that can obtain the best compensation among random-inverse distributions. On the other hand, Figure 8b shows the distribution maps made with the RDPS distribution pattern with the least compensation effect among the random-inverse distributions. Analyzing the dispersion maps shown in Figure 8a, it can be recognized that the amount of accumulated dispersion (i.e., positive values) and the amount of offsetting dispersion (i.e., negative values) is intensively large at the beginning and the end of the same transmission path.

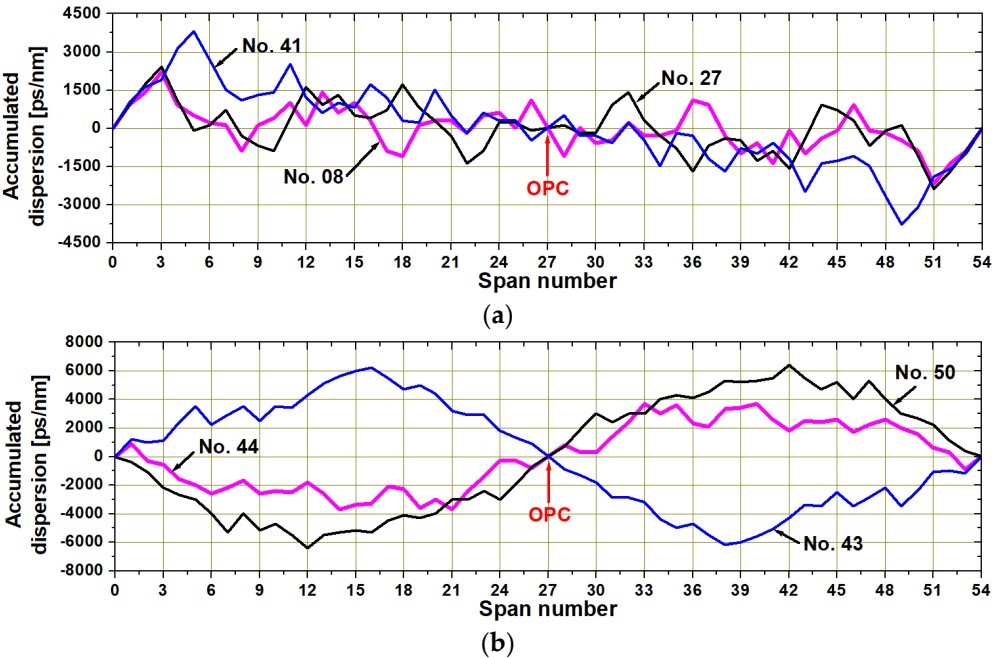

**Figure 8.** The dispersion maps established by random-inverse distribution of RDPSs. (**a**) The best compensation cases, and (**b**) the worst compensation cases.

The optical pulse width in fiber spans in which the accumulated dispersion amount has a positive value is broader than the initial pulse width. As the width of the optical pulse is extended, the intensity of the optical pulse is weakened, and as a result, it is less affected by the nonlinear Kerr effect. However, if the cumulative dispersion is maintained only as positive value in all fiber spans in order to be less affected by the Kerr effect, inter-symbol interference (ISI) occurs due to the dispersion of optical pulses, which deteriorates the transmission quality. Therefore, it is necessary to make the cumulative dispersion amount negative at appropriate intervals. The dispersion maps shown in Figure 8a correspond to the shape following the aforementioned distribution of the accumulated dispersion.

On the other hand, the dispersion maps shown in Figure 8b are also symmetric around the midway OPC but show different aspects from Figure 8a. In particular, the dispersion map generated by pattern number 43 has positive cumulative dispersion in the front part of the former half and negative cumulative dispersion in the rear part of the latter half, like the dispersion maps shown in Figure 8a. However, in this dispersion map, the overall shape of the accumulation and offset of the dispersion in each half link is generally wider than that in Figure 8a.

The conclusion drawn from the analysis of Figure 8 is that effective compensation can result when the RDPS is randomly set to repeat the accumulation and offset of the dispersion in a relatively short period after the positive cumulative dispersion is concentrated at the beginning of the entire link. In addition to this, by setting the RDPS distribution order in the latter half link opposite to that in the former half, effective compensation can be obtained.

## 5. Summary and Consideration for Implementation of Optical Backbone

The MSSI-based DM link proposed in this study can be used as optical backbone link. In this section, we summarize the simulation results and mention the points to be considered when implementing the results of this study into an optical fiber backbone network.

The random distribution of RDPSs in the former half link and the reverse distribution to those of the former half in the latter half link is the most suitable configuration capable of compensating for the distorted WDM signal. This result means that the midway OPC significantly affects the compensation performance compared to the dispersion management. Although the RDPS random distribution of fiber spans can increase the flexibility of link design compared to the conventional DM system, the system performance is deteriorated due to the irregularly shaped dispersion-power profile. However, the random-inverse distribution can make the dispersion-power profile almost symmetrical with respect to the midway OPC; thus, it is possible to satisfy the conditions necessary for compensation of the signal distortion due to not only the chromatic dispersion but also the nonlinear Kerr effect, especially SPM.

In the design stage, by applying MSSI-based DM to the ultra-long haul backbone link, it is common to allocate RDPS for all fiber spans uniformly, but the practical process is not easy because of the various field conditions. That is, in practice, it may be more convenient to allocate the different RDPS to each fiber span. As a result, the random distribution of RDPS is realistic in backbone link design. It is expected that the compensation deterioration derived from the field circumstance can be reduced by making the random distribution of RDPS before and after OPC symmetry.

The all-random distribution is more advantageous than the random-inverse distribution in terms of the practical convenience of RDPS allocation. Although the MSSI-based DM link configured with the all-random distribution cannot good compensate for the distorted WDM signal, it is improved over the random-follow distribution. We confirm that the allowable launch power and effective NRD range can be maintained as much as in the conventional scheme by applying the special random distribution pattern that can approximately symmetrize the dispersion-power profile, even when the WDM signal is transmitted through the MSSI-based DM link in which the RDPSs in both halves are randomly distributed.

The compensation effect of the distorted WDM signal through the link proposed in this study may not be comparable to that of the MSSI-based DM link configured with the artificial distribution of RDPS. This is because all RDPSs are intentionally distributed to set up the symmetric dispersion-power profile with respect to the OPC. However, even with the excellent DM configuration for the compensation, it requires effort to find the optimal dispersion map and not the RDPS combination pattern in the artificial distribution.

## 6. Conclusions

We investigated the compensation of the distorted WDM signal through the MSSI-based DM link configured with the randomly distributed RDPS in the former half link. The RDPS arrangement in the latter half was classified into the cases of following or reversing the random distribution arrangement of those in the former half (random-follow or random-inverse), and the case of random with a different pattern irrespective of the arrangement in the former half section (all-random).

It was confirmed that the compensation effect of the distorted WDM channel was improved overall in the link to which the dispersion map made of random-inverse distribution was applied, among the three cases. This result seems to be because the random-inverse

distribution can establish a more symmetrical distribution map with respect to the midway OPC. In addition, it was confirmed that the dispersion maps generated by the random-inverse distribution are much more advantageous than the other two distributions for the flexibility of the dispersion-managed link design.

The intention of this paper is to provide engineers with a flexible dispersion map configuration without the limitation of RDPS arrangement in designing a dispersion-managed link with the midway OPC, and to provide a sufficient and selectable design margin at these links. The random distribution of the RDPSs proposed in this study, especially the random-inverse distribution, and the NRD margin and power margin in this configuration can be seen as the result that can sufficiently satisfy the purpose of our study compared to the existing links such as the conventional DM link.

However, it requires a lot of effort to randomly arrange the RDPS of the fiber spans consisting of the former half link so that acceptable compensation quality can be derived, and even if such a random pattern is found, another complication may be that the RDPS arrangement in the latter half link has to be reversed to that of the former half.

Nevertheless, we think that the method of randomly distributing the RDPS of the fiber spans of the former half link and artificially constructing the RDPS distribution in the latter half can be considered a new approach to flexibly configuring the dispersion-managed link and increasing the compensation effect at the same time.

Furthermore, as we specified the shape of the dispersion map based on random-inverse distribution that can further improve the compensation performance, it is expected that this work can practically contribute to design and setup of the MSSI-based DM link with the detailed arrangement guidance and the system margin of random RDPS induced in this study.

**Author Contributions:** Conceptualization, J.-P.C. and S.-R.L.; methodology, S.-R.L.; software, S.-R.L.; analysis, J.-P.C. and S.-R.L.; resources, S.-R.L.; data curation, S.-R.L.; writing—original draft preparation, J.-P.C. and S.-R.L.; writing—review and editing, J.-P.C. and S.-R.L.; visualization, S.-R.L.; super-vision, J.-P.C. and S.-R.L.; project administration, S.-R.L. All authors have read and agreed to the published version of the manuscript.

**Funding:** This research received no external funding.

**Institutional Review Board Statement:** Not applicable.

**Informed Consent Statement:** Informed consent was obtained from all subjects involved in the study.

**Data Availability Statement:** The data presented in this study are available on request from the corresponding author. The data are not publicly available due to institutional regulations.

**Conflicts of Interest:** The authors declare no conflict of interest regarding the publication of this paper.

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
