# Peer review of "MSSI-Based Dispersion-Managed Link Configured by Randomly-Distributed RDPS Only in Former Half Section"

_applsci, doi:10.3390/app12188970_

Round 1

Reviewer 1 Report

The paper MSSI-based Dispersion-managed Link Configured by Randomly-distributed RDPS only in Former Half Section is focused on chromatic dispersion and Kerr effect compensation in optical fibers using optical phase conjugator (OPC). The authors provide a technique of using inverse arrangement of optical fiber spans based on their dispersion parameters together with OPC placed in the middle. They analyzed and simulated three different arrangement scenarios in order to provide the optimum conclusion.

First, the length of paper is appropriate, its overall organization is acceptable and the content is technically sound.

The language level throughout the entire paper should be increased as I noticed various different mistakes and problems in the text. I recommend to perform a careful proofreading.

All presented results and their conclusions are based on simulations only. I recommend to perform an experimental verification of proposed conclusions in order to support them. Moreover, I miss more detailed information about simulation and simulator used.

The list of references contains mostly older and not up-to-date references.

According to its introduction, the paper should deal with chromatic dispersion and Kerr effect compensations, however, the rest of the paper is focused on dispersion compensation only and Kerr effect is not mentioned.

Although the conclusion of the paper contains some brief conclusion towards practical implementation of the presented results, I think the paper should contain a section with explanation of the results and their practical implementation in practice.

Reviewer 2 Report

Thank you for submitting your manuscript. This manuscript proposes a dispersion management approach by randomly arranging the RDPS of the former half of the optical link and constructing the latter half link with reversed configuration.  The following questions should be addressed before the publication.

1.      Please provide detailed information on how the simulation is actually carried out in this paper.

2.      Does the author have any experimental results of applying this dispersion management approach to the physical optical link to support the simulation results?

3. Compared with other approaches, what is the complexity of this approach? Will it increase the cost and reduces the performance of the optical link?

4.      The acronym of WDM has been explained twice. Please correct it.

5.      Please correct all the typos in the manuscript. 

Round 2

Reviewer 1 Report

All major suggestions and queries were sufficiently addressed, the paper can be accepted now.